# Targeted vaccination is effective for mpox clade Ib in England despite increased household transmission: Predictions from a modelling study

**Ellen Brooks-Pollock**[1,2☉*], **Leon Danon**[2,3,4☉*]

**1** Population Health Sciences, Bristol Medical School, University of Bristol, Bristol, United Kingdom,
**2** Bristol Vaccine Centre, University of Bristol, Bristol, United Kingdom, **3** School of Engineering Mathematics and Technology, University of Bristol, Bristol, United Kingdom, **4** Jean Golding Institute, University of Bristol, Bristol, United Kingdom

☉ These authors contributed equally to this work.
* Ellen.Brooks-Pollock@bristol.ac.uk (EBP); L.Danon@bristol.ac.uk (LD)

## Abstract

Mpox is an emerging infectious disease with increasing global relevance. In 2024, a new clade, mpox clade Ib, was associated with high household attack rates and case fatality, raising concerns about sustained human-to-human transmission outside endemic areas. We developed and applied an individual-based probabilistic framework for mpox incorporating sexual and household contact patterns using data from the National Survey of Sexual Attitudes and Lifestyles 3 (NATSAL3). Individual and population-level reproduction numbers were estimated using setting-specific secondary attack rates. Vaccination impact was assessed across scenarios varying household attack rates, vaccine effectiveness, and distribution strategies. We predict that fewer than 3 out of 100 individuals generated more than one secondary case of clade Ib in England, with a population reproduction number of 0.69 (95%CI 0.66 – 0.71). Individuals reporting both same-sex and opposite sexual contact disproportionately contributed to transmission potential. Increasing household secondary attack rates led to modest increases in the population reproduction number due to individuals with the highest reproduction numbers having lower than average numbers of household contacts. Targeted vaccination, focused on individuals with higher numbers of sexual contacts, consistently outperformed non-targeted strategies, requiring lower vaccine coverage to achieve control even under elevated household transmission. Vaccine effectiveness against infection and onward transmission critically influenced the success of vaccination programs. Despite higher household transmission risks, targeted vaccination remains an effective strategy for controlling mpox clade Ib in England. Transmission dynamics are strongly shaped by underlying contact structures, emphasizing the importance of network-informed interventions. Rapid, network-informed models can provide valuable early guidance for emerging infectious diseases.

**Data availability statement:** NATSAL-3 data are available from the UK Data Service at https://doi.org/10.5255/UKDA-SN-7799-2. Code repository for all results is found here: https://github.com/bristol-vaccine-centre/MPox-Reckoners. DOI: 10.5281/zenodo.17008814.

**Funding:** EBP acknowledges the support of the National Institute for Health Research Health Protection Research Unit (NIHR HPRU) in Evaluations and Behavioural Science at the University of Bristol. EBP and LD acknowledge the support of the National Institute for Health Research Health Protection Research Focus Award in Immunisation at the University of Bristol. LD is funded by UK Research and Innovation AI programme of the Engineering and Physical Sciences Research Council (EPSRC grant EP/Y028392/1). The funders had no role in study design, data collection and analysis, decision to publish, or preparation of the manuscript.

**Competing interests:** The authors have declared that no competing interests exist.

## Introduction

Mpox is an emerging infectious disease that requires a response that adapts to emerging evidence. Mpox is an Orthopoxvirus, originally known as monkeypox due to its zoonotic origin and closely related to cowpox and smallpox. Although the first human cases were reported in 1970, mpox remained largely endemic to Africa, with sporadic cases reported outside of these regions [1]. In 2022, the identification of a new mpox clade, clade IIb, marked the onset of widespread global transmission and provided the first evidence of sustained human-to-human spread outside of endemic areas.

In 2024, the World Health Organization (WHO) issued a global health alert in response to the emergence of another new clade, clade Ib, the second time mpox has been declared a Public Health Emergency of International Concern in two years, highlighting the persistent threat posed by the virus globally. Clade Ib was associated with increasing numbers of cases, particularly in children, high case fatality rates, and high household attack rates [2]. Since 2024, it has caused over 30,000 reported cases, mostly in the Democratic Republic of Congo with evidence of community transmission in neighbouring countries, and travel-associated sporadic cases in Asia and the Middle East, North and South America, and Europe [3]. Due to its characteristics, clade Ib was classified as a High Consequence Infectious Disease in the UK, whereas previous clades were not [4]. Whilst the immediate risk of sustained community transmission of clade Ib outside endemic regions has diminished, characterising the importance of household transmission on population-level spread remains important.

Mpox infection commonly manifests with skin lesions, reported in more than 95% of patients in the 2022 outbreak [5]. Other symptoms include fever, aches, pains and other non-specific symptoms [6]. Infection is mostly mild and self-limiting but can be severe in immunocompromised individuals [6]. Transmission occurs through direct skin-to-skin contact, with secondary attack rates higher for sexual compared to non-sexual contacts and household compared to non-household contacts [7].

The structure of contact networks plays a critical role in the transmission of infectious diseases [8]. Besides the number of contacts an individual has, the arrangement of those contacts, such as the degree-degree correlation, can accelerate or mitigate outbreaks [9]. High assortativity gives rise to core groups that can drive outbreaks and mean that targeted interventions can be highly effective. The mpox clade IIb outbreak in Europe in 2022 disproportionately affected Gay/Bisexual and Men-who-have-sex-with-men (GBMSM) [10–12]. Due to this heterogeneity in the transmission network, targeted interventions such as targeted vaccination and contact tracing [13] were highly effective at controlling transmission [14,15], along with behaviour change [16,17].

There remains considerable uncertainty about the infectiousness of clade Ib and the effectiveness of vaccines against infection and transmission [18]. In this study, we examine the transmission of mpox clade Ib in England through sexual and household contacts and assess the potential for vaccination to control transmission

under uncertainty. Using a rapid evaluation framework that links contact patterns to the reproduction number, our findings emphasize the importance of network structure for the sustained transmission of close contact infections such as mpox. The framework we develop is open-source and can be adapted for use in other contexts.

## Methods

### Modelling approach

We used an individual-based disease model based on our previous work [19,20]. For each individual, we calculate an individual reproduction number, $R_i$, given by equation (1):

$$R_i = \sum_{k=1}^{3} [SAR]_k n_{k,i},$$

Where $k = 1, 2, 3$ corresponds to three types of physical contact: household contact, heterosexual sexual contact and same-sex sexual contact; $[SAR]_k$ is the setting-specific secondary attack rate (proportion of contacts that result in secondary infection); and $n_{k,i}$ is the number of contacts per setting for individual $i$. Although reported secondary attack rates include secondary, tertiary and subsequent infections, we use the reported secondary attack rate as a proxy for the transmission probability because we do not model transmission events between contacts of egos in this framework.

The population-level reproduction number is derived from the individual reproduction numbers, assuming proportionate mixing between individuals, i.e., that the probability of contacting individual $i$ is proportional to their number of contacts over the total number of contacts in the population, $\frac{R_i}{\sum_i w_i R_i}$. The population-level reproduction number scales with the square of the individual-level reproduction numbers:

$$R_t = \frac{\sum_i w_i (R_i)^2}{\sum_i w_i R_i},$$

where $w_i$ is the individual weighting and $\sum_i w_i = 1$.

We extended the model to include the impact of vaccination by modifying the risk of infection and the risk of onward transmission. We denote the reduction in risk of infection from vaccination as $v_I$, and the reduction in infectiousness associated with vaccination as $v_T$. In an otherwise fully susceptible population, a vaccinated person's individual reproduction number becomes:

$$R_i{}^{vac} = (1 - v_T) \sum_{k=1}^{3} [SAR]_k n_{k,i}.$$

The probability that a vaccinated individual is infected becomes

$$\frac{(1 - v_I) w_i n_i}{\sum_j w_j n_j}.$$

The population-level reproduction number is

$$R_t^{vac} \sim \sum_{i=1}^{k} p_i w_i (1 - v_T)(1 - v_I) n_i c_i + \sum_{i=k+1}^{2k} (1 - p_i) w_i n_i c_i$$

where $p_i$ is the proportion of group $i$ that is vaccinated. Full details in the Supplementary Information.

To simulate non-targeted vaccination, we assume that all individuals have an equal probability of being vaccinated, i.e., $p_i \equiv p$, for all $i$. For targeted vaccination, we take $p_i \propto p^{n_i}$, where $n_i$ is the total number of contacts of individual $i$, and ensuring that $\sum_i w_i p_i = p$. We repeat each analysis for vaccine effectiveness estimates for pre- and post-exposure vaccination effectiveness.

### Data and parameter estimates

Model parameters are given in Table 1. We used data from the National Survey of Sexual Attitudes and Lifestyles 3 (NATSAL3), a representative sample of individuals aged 16–74 years of age in Great Britain [21,22]. We included all individuals and extracted the following variables: individual weighting, age, household size, number of heterosexual partnerships in the previous year and the previous five years, and number of same sex partnerships in the previous year and previous five years. Analyses were conducted on the entire dataset, as well as the subgroup of individuals who reported at least one same-sex sexual relationships within the five years.

We used attack rates estimated from clade IIb transmission in the 2022 outbreaks in England [7], comparing the resulting reproduction number estimates against existing published values [23,24]. We included additional household transmission to explore the potential impact of clade Ib in England. We used household attack rates ranging from 1.3% reported in [7] to 15% [25], further exploring the impact of hypothetical extremes of up to 50%.

Pre-exposure vaccination with a third-generation smallpox vaccine has been shown to be highly effective [26,27]. The effectiveness against becoming a case was estimated for England as 82% (95% CI 74%–88%), with no significant difference between one and two doses (Charles et al., 2024). Post-exposure vaccination is also used, not to prevent infection [29], but to reduce symptoms [30]. For post-exposure vaccination, the effectiveness is estimated to be lower; based on time since last exposure, a study of cases in New York estimated an effectiveness of 19% (95% CI -54%, 57%) [29]. For pre-exposure vaccination, we assume that both $v_T, v_I > 0$, whereas for post-exposure vaccination we assume that $v_T = 0$ but $v_I > 0$.

### Results

For clade IIb, the population reproduction number was estimated to be 0.47 (95% CI 0.40 – 0.54) for the entire population, with less than 1.5% of individuals estimated to have an individual reproduction number greater than one. The maximum individual reproduction number was 6.6. Individuals who reported mainly, but not exclusively, same-sex sexual contact in the previous 5 years have on average, a greater number of sexual contacts which translates to higher individual

**Table 1. Model parameter symbols, interpretation, values and sources.**

| Parameter | Interpretation | Values (uncertainty) | References |
|---|---|---|---|
| $n_{k,i}$ | Number of contacts for individual $i$ by setting (household contacts, same-sex sexual contacts, opposite-sex sexual contacts). | Varies by individual | [21,22] |
| $[SAR]_k$ | Secondary attack rate by setting | Household (non-sexual) contacts = 1.3%<br>Same-sex sexual contacts = 12%<br>Opposite-sex sexual contacts = 12% | [7] |
| $w_i$ | Individual weighting to capture NATSAL sampling | Varies by individual | [21,22] |
| $v_T$ | Reduction in infectiousness due to vaccination | Pre-exposure: 50% | [30] |
| | | Post-exposure: 0% | |
| $v_I$ | Reduction in the risk of disease due to vaccination | Pre-exposure: 82% (95% CI 74%, 88%) | [28] |
| | | Post-exposure: 19% (95% CI -54%, 57%) | [29] |
| $p_i$ | Probability that individual $i$ is vaccinated and proportion of contacts that are vaccinated | Varied within vaccination scenarios between 0 and 1 | |

reproduction numbers. Whilst 3.5% of individuals reported a same sex sexual contact in the previous 5 years, they represented 16% of individuals with a reproduction number greater than one. This means that in this subgroup, the corresponding population reproduction number is greater, 1.2 (95%CI 0.62 – 1.60). Household close contacts contribute minimally to the reproduction numbers, due to the low household attack rates. Individuals reporting same-sex contact are associated with smaller household sizes.

Increasing household attack rates to values measured for sexual contacts (approximately 13%), leads to modest increases in the population reproduction number to 0.69 (95%CI 0.66 – 0.71). Under this scenario of increased household attack rates, 3 in 100 individuals are predicted to generate more than one secondary case, and individuals reporting both same-sex and opposite-sex sexual contacts disproportionately contribute to transmission potential (Fig 1).

We explored the impact of increasing household attack rates further and the ability to control transmission through vaccination in individuals reporting at least one same-sex contact (Fig 2). The 2022 clade IIb scenario is represented in the leftmost panel with a household attack rate of ~0.13%. For moderate increases in the household attack rate (up to 20%) there is a limited impact on the reproduction number, demonstrated by the maintenance of low vaccination thresholds for targeted vaccination (Fig 2 bottom panel). With vaccination, higher uptake rates are required to control transmission for both non-targeted and targeted vaccination programmes, with targeted vaccination always being more effective.

For all scenarios explored, targeting vaccination by risk group (in this case defined by numbers of contacts) is highly impactful, often rapidly reducing the population reproduction number to less than one (green shaded regions). Even with high household attack rates, less than half of individuals need to be vaccinated. In contrast with a non-targeted approach to vaccination (red shaded region) where two thirds of individuals would need to be vaccinated to achieve similar reductions. For high household attack rates the relative benefit of targeted versus non-targeted vaccination is reduced, although vaccinating the highest risk individuals remains the most beneficial approach (Fig 2).

In addition to vaccine uptake and distribution, the vaccine effectiveness and mechanism of vaccine action have a major impact on the success of any vaccine programme (Fig 3). We explored the impact of vaccines preventing disease (Fig 3 rows) and vaccines preventing transmission (Fig 3 columns) for non-targeted and targeted vaccination. Disease-preventing vaccines have a different impact to transmission-blocking vaccines, and both effects interact with vaccine uptake and vaccine distribution.

For low vaccine effectiveness against blocking disease and blocking transmission (less than a 20% reduction in disease or transmission with vaccination), both non-targeted and targeted vaccination have limited impact, and did not control transmission in our analyses. For more effective vaccines, targeted vaccination is always more effective than non-targeted vaccination, particularly for vaccines that effectively block transmission.

For vaccines that provide effective direct protection against disease, there is a linear association between the proportion of individuals receiving a non-targeted vaccine and the reproduction number (red shaded regions). Targeted vaccination has a non-linear benefit, with rapid reductions in the reproduction number for low uptake numbers (green shaded region in panels).

## Discussion

Our findings show that even large increases in mpox household secondary attack rate would likely have a small impact on transmission at the population scale. We found targeted interventions to be highly effective, although increases in household transmission diminish the impact of targeted approaches. Consequently, close monitoring of attack rates in non-sexual contacts is essential for assessing transmission risks in non-endemic settings. This analysis highlights the benefit of targeting vaccination towards high-risk individuals to effectively control mpox transmission. However, identifying high-risk individuals in practice can be challenging; vaccination programmes use a broader definition of high-risk to create a workable policy, such as attendance at sexual health clinics.

PLOS Global Public Health

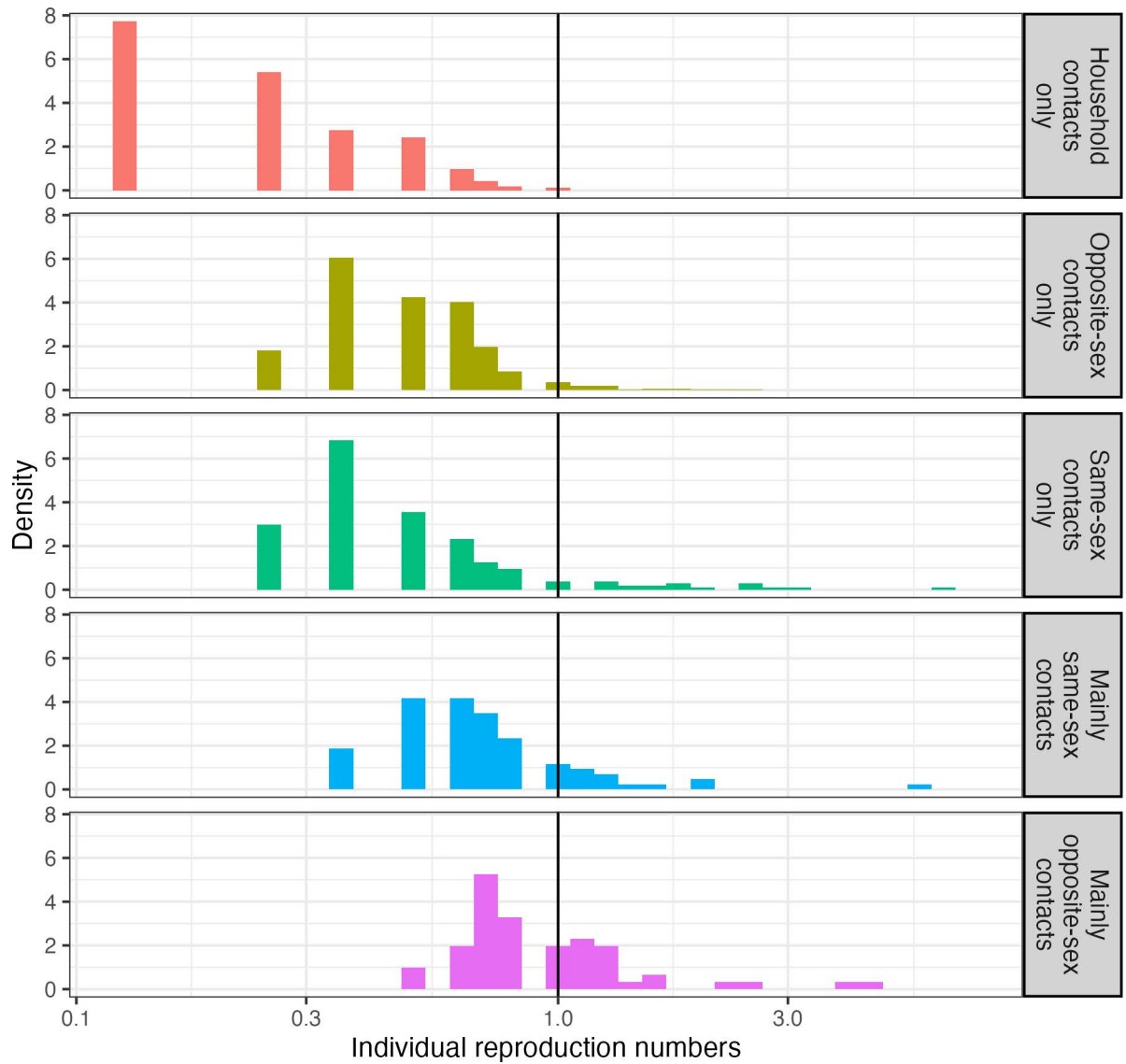

**Fig 1. Estimated individual reproduction numbers for mpox clade Ib for NATSAL3 respondents, separated by types of reported contact.** The vertical black line marks where the individual reproduction number equals 1 – individuals to the left of the black line are predicted to generate less than one secondary case on average, individuals to the right of the line would be predicted to infect more than one other individual on average.

Heterogeneity is a key driver of mpox transmission in endemic settings. While we estimate a population-level reproduction number of less than one, meaning that widespread transmission is not sustained, transmission among core groups could be sustained, hence the need for targeted interventions. Our results for clade IIb are consistent with previous modelling work based on NATSAL [15]; and here we have shown that increased household transmission reduces but does not eliminate the importance of core groups, and these results are robust to uncertainty in household attack rates.

In this study, we used data from the National Surveys of Sexual Attitudes and Lifestyles 3 (NATSAL3) to inform the transmission model. NATSAL provides valuable sexual behavioural data, spanning multiple years [15]. All surveys, including NATSAL, have variable representativeness and reporting biases, which may particularly affect some

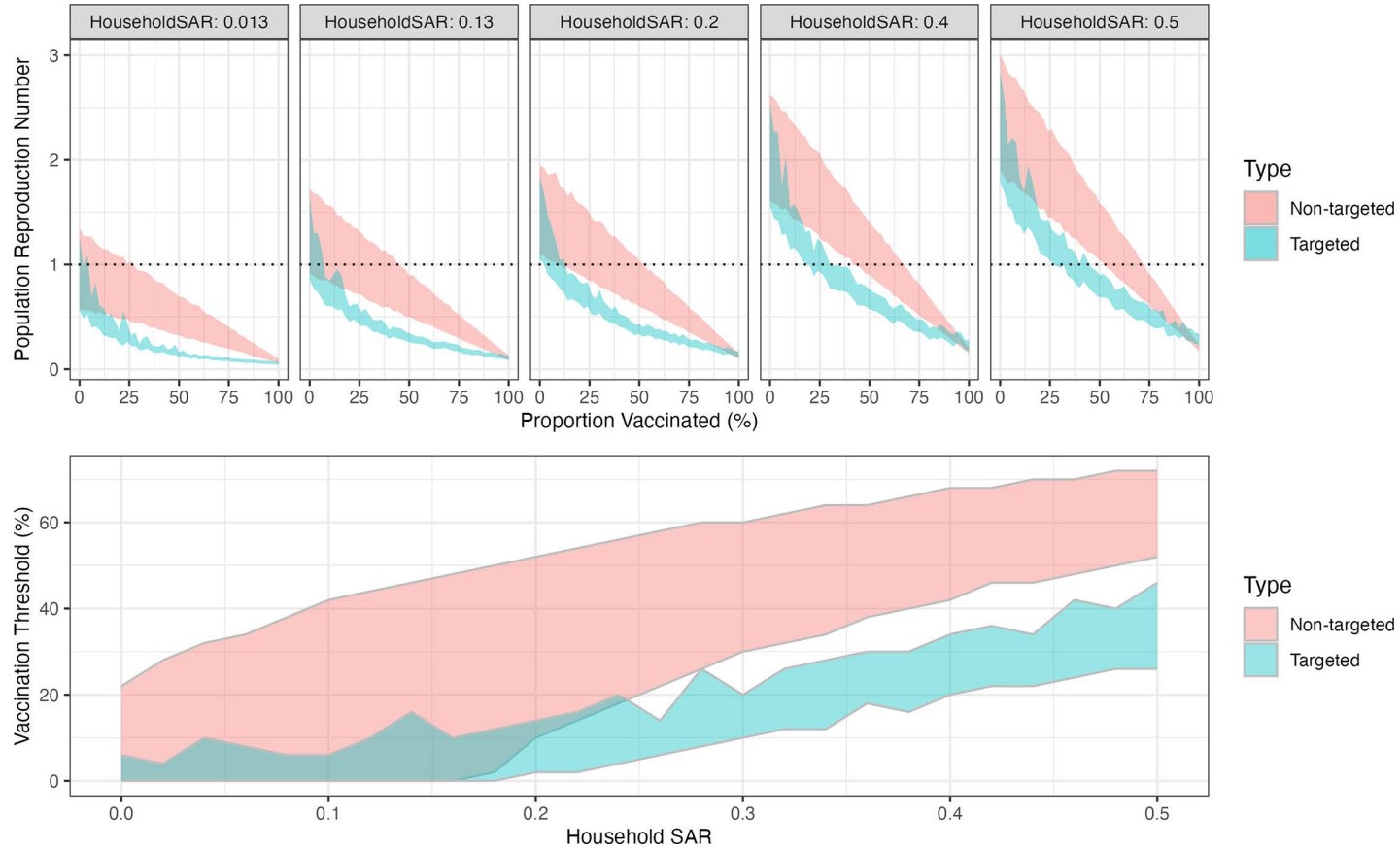

**Fig 2. Population Reproduction number and vaccination thresholds.** Top: Population reproduction numbers as a function of the percentage of the population vaccinated in individuals reporting same-sex sexual contacts in NATSAL3. Each panel represents a household secondary attack rate value; vertical lines show the threshold at which the reproduction number crosses 1; horizontal dotted line show R = 1. Bottom: Vaccination threshold as a function of the household secondary attack rate. Colours in all panels denote the 95% confidence interval on the mean by type of vaccination strategy employed.

questions on sensitive topics. Due to sampling by postcodes, NATSAL does not include people who are homeless or living in institutions [22], however measured biases have remained largely consistent across surveys [31]. Here, the inclusion of household size data, in addition to sexual behaviour, enabled us to capture non-sexual close contact transmission in the model. In future, inclusion of social contact data as part of NATSAL would further enhance the power of models, particularly for infectious diseases with multiple transmission routes. The integration of clinical data would allow direct calculation of secondary attack rates and vaccine effectiveness, rather than relying on external estimates.

Our results are dependent on the contact and household structure in the UK, therefore are not necessarily applicable to other settings. In particular, the lack of association between household size and number of sexual contacts (i.e., individuals reporting high numbers of sexual contacts reported small household sizes), may not apply elsewhere. Other limitations of our approach include that we do not capture higher-order network effects on transmission, such as clustering, household structures and groups of individuals. Our approach does not model epidemic trajectories; therefore we are not able to explore dynamic or adaptive control measures. Furthermore, our approach does not capture the impact of building immunity on transmission or the effectiveness of interventions.

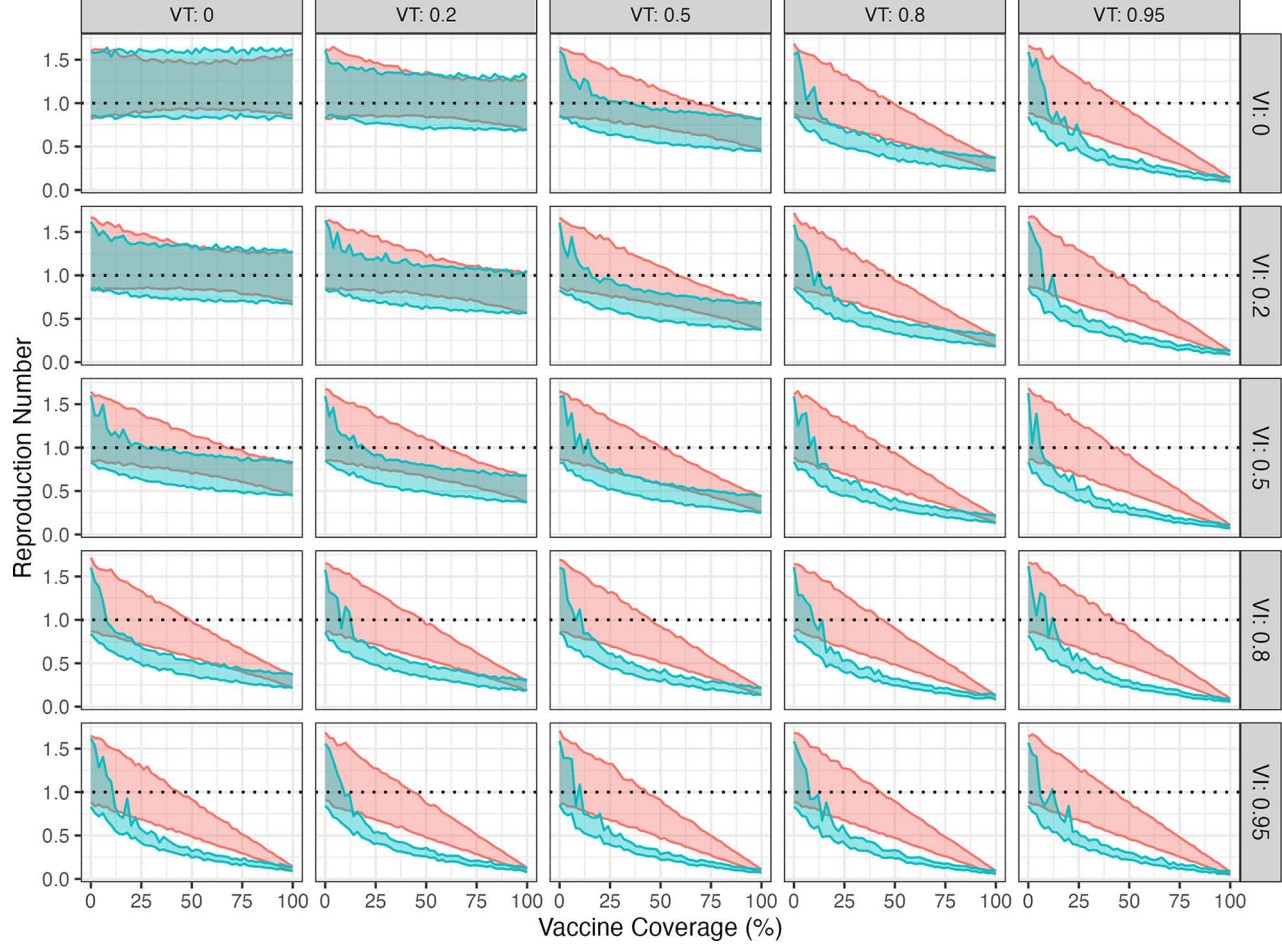

**Fig 3. The relationship between vaccine mechanism of action, uptake and vaccine effectiveness for the subpopulation of individuals who report same-sex sexual contacts in NATSAL3 for clade Ib parameters (household attack rate=13%).** VT is the vaccine effectiveness against transmission (**VT = 0** *is no impact,* **VT = 0.95** *means vaccinated individuals are 5% as infectious as non-vaccinated individuals). VI is the vaccine effectiveness against infection. Red coloured regions represent 95% confidence intervals on the mean for non-targeted vaccination and green shaded regions are targeted vaccination.*

This modelling work was conducted in 2024 following the first cases of mpox clade Ib in England. The modelling framework balances data-intensive and computationally demanding individual-based models and simpler compartmental frameworks, while still capturing the non-linearity inherent in disease transmission. Non-dynamic, at-a-glance modelling approaches, such as ours, provide an early and rapid assessment of transmission potential and control options and associated uncertainty. This enables decision-makers to consider a range of options before commissioning more detailed models that allow for a deeper exploration of granular control measures. Such low resource models have the potential to be impactful in settings with limited data and resources.

In conclusion, this study provides insight into the transmission of mpox in the UK and the potential for control in the context of increased household transmission. Our findings highlight that targeted interventions, especially when directed towards high-risk populations, remain a highly effective strategy for limiting transmission. As the situation evolves, continuous monitoring of transmission in non-sexual contacts, vaccination uptake and immunity will be essential for ensuring an effective public health response.

## Acknowledgments

We would like to thank Rajeka Lazarus and UKHSA colleagues for comments on early versions of this work.

## Author contributions

**Conceptualization:** Ellen Brooks-Pollock, Leon Danon.

**Data curation:** Ellen Brooks-Pollock, Leon Danon.

**Formal analysis:** Ellen Brooks-Pollock, Leon Danon.

**Funding acquisition:** Ellen Brooks-Pollock, Leon Danon.

**Investigation:** Ellen Brooks-Pollock.

**Methodology:** Ellen Brooks-Pollock, Leon Danon.

**Resources:** Leon Danon.

**Software:** Ellen Brooks-Pollock, Leon Danon.

**Validation:** Ellen Brooks-Pollock, Leon Danon.

**Visualization:** Ellen Brooks-Pollock, Leon Danon.

**Writing – original draft:** Ellen Brooks-Pollock.

**Writing – review & editing:** Ellen Brooks-Pollock, Leon Danon.

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
