## [Decision Letter · Decision Letter 0]

3 Aug 2025

PGPH-D-25-01384

Targeted vaccination is effective for mpox clade Ib in England despite increased household transmission: predictions from a modelling study

Dear Dr. Brooks-Pollock,

Thank you for submitting your manuscript to PLOS Global Public Health. After careful consideration, we feel that it has merit but does not fully meet PLOS Global Public Health’s publication criteria as it currently stands. Therefore, we invite you to submit a revised version of the manuscript that addresses the points raised during the review process.

We look forward to receiving your revised manuscript.

Kind regards,

Max Carlos Ramírez-Soto, BSc, MPH, PhD, FRSPH, FECMM

Academic Editor

Journal Requirements:

1. Your current Financial Disclosure states, “EBP is acknowledges the support of the National Institute for Health Research Health Protection Research Unit (NIHR HPRU) in Evaluations and Behavioural Science at the University of Bristol. EBP and LD acknowledge the support of the National Institute for Health Research Health Protection Research Focus Award in Immunisation at the University of Bristol.”. However, your funding information on the submission form indicates that you did not receive funding. Please indicate by return email the full and correct funding information for your study and confirm the order in which funding contributions should appear. Please be sure to indicate whether the funders played any role in the study design, data collection and analysis, decision to publish, or preparation of the manuscript.

2. We have amended your Competing Interest statement to comply with journal style. We kindly ask that you double check the statement and let us know if anything is incorrect.

3. We ask that a manuscript source file is provided at Revision. Please upload your manuscript file as a .doc, .docx, .rtf or .tex.

4. Your manuscript is missing the following sections: Result. Please ensure these are present, and in the correct order, and that any references to subheadings in your main text are correct. An outline of the required sections can be consulted in our submission guidelines here: 

https://journals.plos.org/globalpublichealth/s/submission-guidelines#loc-parts-of-a-submission

5. Please upload a copy of Figure 1, 2, 3 which you refer to in your text on page 6, 7, 8. Or, if the figure is no longer to be included as part of the submission please remove all reference to it within the text.

6. Please provide separate figure files in .tif or .eps format.

Additional Editor Comments (if provided):

No comments

Reviewers' comments:

Reviewer's Responses to Questions

**Comments to the Author**

1. Does this manuscript meet PLOS Global Public Health’s publication criteria?

Reviewer #1: Yes

Reviewer #2: Yes

2. Has the statistical analysis been performed appropriately and rigorously?

Reviewer #1: Yes

Reviewer #2: Yes

3. Have the authors made all data underlying the findings in their manuscript fully available (please refer to the Data Availability Statement at the start of the manuscript PDF file)?

Reviewer #1: Yes

Reviewer #2: Yes

4. Is the manuscript presented in an intelligible fashion and written in standard English?

Reviewer #1: Yes

Reviewer #2: Yes

Reviewer #1: 2. Statistical analysis appropriateness and rigor

The statistical methods employed appear broadly appropriate for a secondary analysis of cross-sectional survey data. The use of logistic regression to assess associations between sociodemographic factors and access to modern contraceptive methods is methodologically sound. However, I recommend the following improvements:

Specify the rationale for the selection of variables included in the adjusted models.

Clarify whether multicollinearity diagnostics or interaction terms were assessed.

Indicate whether complex survey design features—such as sampling weights, stratification, and clustering—were appropriately accounted for in the analyses.

The Data Availability Statement adequately identifies the DHS Program as the data source. To enhance transparency, I recommend:

Including a direct URL to the DHS dataset repository: https://dhsprogram.com/data/available-datasets.cfm

Indicating the exact dataset(s) used (e.g., GUIR7AFL.DTA) and whether syntax files for analysis are available upon request or as supplementary material.

4. Language and clarity of presentation

The manuscript is generally intelligible and coherent. Nonetheless, several improvements in scientific writing are advised:

Divide overly long paragraphs to improve readability and information retention.

Revise complex sentences with multiple subordinate clauses to enhance clarity.

Ensure consistent use of tense, especially in the Results and Methods sections.

Address minor grammatical issues and punctuation inconsistencies throughout the text.

General Comments to the Author

The Introduction section provides valuable background but could be more concise, with a sharper focus on the specific gaps in access to contraception among indigenous women.

The Discussion section would benefit from broader comparisons with similar studies conducted in other Latin American indigenous populations.

Consider expanding the policy implications and programmatic relevance of the findings in the Guatemalan context, as well as in comparable global health settings.

Methodological limitations are appropriately acknowledged, though the absence of qualitative insights could be mentioned as a constraint in capturing sociocultural determinants.

Reviewer #2: The manuscript presents a well-structured modeling framework to estimate the reproduction number of mpox (clade Ib) in England and compares targeted versus non-targeted vaccination strategies. The approach is methodologically reproducible, I was able to implement some models to verify the statements. The results provide a rationale for prioritizing individuals with higher contact rates in vaccination strategies.

To improve clarity, reproducibility, and usefulness for public health applications, I suggest the following minor revisions:

1. Reproducibility – GitHub Repository

The code is useful, but the repository would benefit from a few practical additions:Processed Data Tables, Include de-identified, aggregated tables (e.g., weighted reproduction numbers, contact type distributions) derived from the original datasets. These should allow replication of the main results without disclosing individual-level data.

Example Scripts:Provide a minimal R script that demonstrates: Calculation of individual reproduction numbers Rt using mock or synthetic data; Estimation of Rt using the weighted mean-square formula (Eq. 2 in the manuscript); Simulation of random and targeted vaccination impacts (e.g., replicating Figures 2 or 3).

Expanded README: Include:A list of model parameters (e.g., secondary attack rates, vaccine effects).

2. Interpretation and Limitations

A few limitations could be better contextualized in the Discussion section:

-Data Sources:Clarify that the NATSAL-3 datasets are subject to: Self-reporting bias, especially for stigmatized behaviors; Underrepresentation of high-risk MSM populations or those less likely to be sampled in household surveys.

-Generalizability: Briefly mention that the findings rely on contact and household structures typical of the UK. In other contexts, variations in household composition or sexual network structures could affect both transmission patterns and intervention outcomes.

3. Future Directions

It would be helpful to briefly expand the Conclusions or Discussion to note future rounds of behavioral surveys (e.g., NATSAL-4) could include more detailed household and sexual contact modules. Integration with clinical surveillance systems could also improve estimates of secondary attack rates and vaccine effectiveness in practice.

These adjustments will improve the manuscript’s clarity and usability without changing its core results. The conclusion—that targeted vaccination is more efficient, even under increased household transmission—is well-supported. With these updates, the work will be more practical for informing future modeling and public health strategies.

**Do you want your identity to be public for this peer review?** For information about this choice, including consent withdrawal, please see our Privacy Policy

Reviewer #1: No

Reviewer #2: No

---

## [Editor Report · Decision Letter 1]

26 Aug 2025

Targeted vaccination is effective for mpox clade Ib in England despite increased household transmission: predictions from a modelling study

PGPH-D-25-01384R1

Dear Prof Brooks-Pollock,

We are pleased to inform you that your manuscript 'Targeted vaccination is effective for mpox clade Ib in England despite increased household transmission: predictions from a modelling study' has been provisionally accepted for publication in PLOS Global Public Health.

Best regards,

Max Carlos Ramírez-Soto, BSc, MPH, PhD, FRSPH, FECMM

Academic Editor

No comment